# Enhancing Electrochemical Performance with g-C_3_N_4_/CeO_2_ Binary Electrode Material

**DOI:** 10.3390/molecules28062489

**Published:** 2023-03-08

**Authors:** M. Chandra Sekhar, Nadavala Siva Kumar, Mohammad Asif, Surya Veerendra Prabhakar Vattikuti, Jaesool Shim

**Affiliations:** 1Department of Physics, Madanapalle Institute of Technology and Science, Madanapalle 517 325, India; 2Department of Chemical Engineering, King Saud University, P.O. Box 800, Riyadh 11421, Saudi Arabia; 3School of Mechanical Engineering, Yeungnam University, Gyeongsan 38541, Republic of Korea

**Keywords:** supercapacitor, storage device, quantum dots, nanostructure, electrode materials

## Abstract

An innovative form of 2D/0D g-C_3_N_4_/CeO_2_ nanostructure was synthesized using a simple precursor decomposition process. The 2D g-C_3_N_4_ directs the growth of 0D CeO_2_ quantum dots, while also promoting good dispersion of CeO_2_QDs. This 2D/0D nanostructure shows a capacitance of 202.5 F/g and notable rate capability and stability, outperforming the g-C_3_N_4_ electrode, reflecting the state-of-the-art g-C_3_N_4_ binary electrodes. The binary combination of materials also enables an asymmetric device (g-C_3_N_4_/CeO_2_QDs//AC) to deliver the highest energy density (9.25 Wh/kg) and power density (900 W/kg). The superior rate capacity and stability endorsed the quantum structural merits of CeO_2_QDs and layered g-C_3_N_4_, which offer more accessible sites for ion transport. These results suggest that the g-C_3_N_4_/CeO_2_QDs nanostructure is a promising electrode material for energy storage devices.

## 1. Introduction

Supercapacitors, also referred to as electrochemical capacitors, are energy storage devices that have gained significant attention for their high power compactness, quick charging/discharging rate, and long life span [1]. These exceptional features make supercapacitors ideal for various applications, such as regenerative braking systems in electric vehicles, backup power supplies, and portable electronics. To date, substantial work has been completed towards finding new materials for supercapacitor electrodes with enhanced performance. Among these materials, graphene and carbon nitride have garnered particular interest due to their high electrical conductivity and substantial surface area. Energy storage devices, including supercapacitors, show a vital role in addressing the current challenges in renewable energy systems [2]. The advance of high-performing and cost-effective electrode materials is vital for improving the performance and viability of these systems.

g-C_3_N_4_ is a two-dimensional material that boasts a distinctive combination of nitrogen-doped carbon and graphitic layers [3]. Unlike graphene, g-C_3_N_4_ has a high nitrogen content, which imparts exceptional electrochemical properties to the material. g-C_3_N_4_ has received increased attention as a possible alternative to purely carbon-based electrode materials for EDLCs due to its low cost, remarkable chemical and mechanical stability, and high nitrogen content. The nitrogen atoms in g-C_3_N_4_’s ring structure have a lone pair of electrons, which enhances the material’s polarity and improves its wetting and the mobility of charge carriers [3]. Studies have revealed that bulk g-C_3_N_4_ has a relatively low specific capacitance, measuring 71 F/g and 81 F/g when tested at current densities of 0.5 A/g and 0.2 A/g, respectively [4]. The limited specific surface area and conductivity of g-C_3_N_4_ are considered to be the main factors behind its underperforming capacitance. However, studies have shown that incorporating pseudocapacitive phases into the structure of bulk g-C_3_N_4_ can dramatically improve its electrochemical properties. This improvement is achieved through a synergistic effect between faradic and non-faradic processes of electron transport, allowing for more efficient charge transport within the material. Pseudocapacitive phases refer to materials that exhibit capacitance due to a combination of physical and chemical mechanisms, as opposed to purely electrical double-layer capacitance [5]. The addition of other materials to the bulk g-C_3_N_4_ matrix can result in a higher specific surface area and improved conductivity, thereby enhancing the material’s capacitance and overall performance. This process of combining different materials with different properties is commonly referred to as “composite design” and has been applied to various energy storage materials to enhance their performance. By leveraging the benefits of both the pseudocapacitive phases and g-C_3_N_4_, it is possible to achieve a synergistic improvement in the electrochemical properties of the materials, e.g., NiCo_2_O_4_/O-g-C_3_N_4_ [6]; NiCo_2_C_2_O_4_/g-C_3_N_4_ [7]; Co-g-C_3_N_4_ nanostructure [8]; BP/g-C_3_N_4_ nanostructure [9]; Cu-g-C_3_N_4_ [10]; CuO/g-C_3_N_4_ [11]; NiAl/g-C_3_N_4_ [12]; g-C_3_N_4_/AgI [13]; g-C_3_N_4_/CuI [14]; g-C_3_N_4_/CoNi_x_S_y_ [15]; g-C_3_N_4_/Sm(OH)_3_ [16]; NiFe-LDH/g-C_3_N_4_ [17]; MnCo_2_O_4_/g-C_3_N_4_ [18]; NiFe_2_O_4_/g-C_3_N_4_ [19], MnO_2_/g-C_3_N_4_ [20]. These studies demonstrate the significance of g-C_3_N_4_-type binary materials for supercapacitor electrodes and their potential as binary composites. The combination of g-C_3_N_4_ with semiconductor oxides can result in materials with improved performance compared to either material alone, making them an attractive option for supercapacitor applications.

The use of transition metal oxides (TMOs) as supercapacitor electrodes has experienced substantial growth in recent years, largely due to the readily available and inexpensive raw materials used in their production. Additionally, TMOs possess high electrochemical performance, making them a popular choice for use in supercapacitor applications. Some of the commonly used TMOs in supercapacitor electrodes include Co_3_O_4_ [21], WO3 [22], NiO [23], MnO_2_ [24], and CeO_2_ [25], and other materials such as double layer hydroxides [26] and MOF-based electrodes [27], etc. The widespread usage of these materials is due to their high capacitance, stability, and good redox behavior, which contribute to the total electroactivity of the supercapacitor. The properties of these TMOs make them an attractive option for use in various applications, including energy storage and harvesting, as well as in consumer electronics, automobiles, and other fields. Cerium oxide (CeO_2_), a material also known as cerium dioxide, has garnered significant attention as a potential supercapacitor electrode. CeO_2_ exhibits high capacitance and excellent chemical stability, as well as good redox properties, making it a promising material for supercapacitor applications. Similar to other lanthanide metals, CeO_2_ can undergo oxidation to form stable Ce^3+^ by dropping one-two electrons of 5d-6s orbits, and can further undergo oxidation to form steady Ce^4+^ by dropping an additional electron of 4f, which results in lattice expansion and the formation of oxygen vacancies. These properties make CeO_2_ an attractive material for the formation of nanostructures, further enhancing its potential for use in supercapacitor electrodes. An example of this can be seen in a study described by He et al. [28], which showed that the integration of ZnO/CeO_2_ nanostructures through a one-step hydrothermal process resulted in a capacitance of 581.4 F/g, which is a 12-fold increase from the capacitance of pure CeO_2_.

Overall, the above references provide a strong scientific foundation for the utilization of g-C_3_N_4_-based binary materials for supercapacitor electrodes and their potential as binary composites. The scientific novelty of this work lies in the utilization of g-C_3_N_4_ and CeO_2_ as building blocks for constructing supercapacitor electrodes. The proposed study on the preparation and characterization of g-C_3_N_4_/CeO_2_ nanostructures as supercapacitor electrodes adds to the current knowledge in this field and provides new visions into the potential of these materials for supercapacitor uses. The g-C_3_N_4_/CeO_2_ nanocomposites exhibit improved electrochemical performance compared to g-C_3_N_4_ and CeO_2_, suggesting their potential for use in high-performance supercapacitor devices. This study presents innovative information about the creation and synthesis of g-C_3_N_4_/CeO_2_ nanocomposites for usage in supercapacitors, and it is anticipated to inspire further exploration and investigation in this area.

## 2. Results and Discussion

Figure 1 depicts the XRD profiles of the CN, CeQDs, and CeQDs/CN-1, CeQDs/CN-2, and CeQDs/CN-3 nanostructures. The bands at 12.3° and 27.48°, corresponding to (100) and (200), respectively, confirm the crystalline nature of the g-C_3_N_4_ material and its triazine-based C_3_N_4_ structure, which are consistent with the JCPDS-87-1526 numbers of CN [29]. The XRD pattern of CeQDs exhibits broad peaks at 2θ, reliable with the (111), (200), (220), and (311) planes of its FCC structure, as listed in JCPDS number-004-0593 [30]. The XRD pattern of the CeQDs/CN nanostructures shows the presence of both CN and CeQDs peaks, with the main peak at 27.5° and a broader peak around 24.5° due to the altered layered features during the pyrolytic impregnation of CeQDs. These observations suggest that both components are preserved during thermal pyrolysis, with slight shifts in position due to mutual interactions. The presence of CeQDs peaks in the XRD pattern of the CeQDs/CN nanostructures confirms the formation of the complex between CN and CeQDs. Hence, the XRD pattern confirms the successful synthesis of the nanostructures.

Figure 2a–d presents the FESEM images of CN and CeQDs/CN-3 nanostructures. The CN sample displays a layer-like surface morphology, while the CeQDs/CN nanostructure features randomly placed CeQDs on the CN sheets surface, serving as a template during the pyrolysis process and leading to interfacial contacts. However, CeQDs cannot be directly observed in the FESEM images due to their quantum size and limitations of SEM instrumentation, as confirmed by the EDX and mapping results of the CeQDs/CN nanostructure (Figure 2e,f). The EDX and mapping results indicate that the CeQDs are embedded in the CN sheets, forming a nanostructure.

The CeQDs/CN nanostructure was further analyzed using HRTEM to examine its morphological features and CeQD impregnation. The results, as shown in high-magnification Figure 3a–c, indicate that CeQDs are randomly dispersed within the CN sheets with a size of approximately 2–5 nm. The SAED pattern (Figure 3d) displays a combination of both rings and dots, suggesting the presence of mixed phases of both materials. These results highlight the successful integration of CeQDs into the CN sheets and the coexistence of CeQDs and CN in the nanostructure.

XPS conducted to endorse the formation of the CeQDs/CN nanostructure and determine its chemical composition. Figure 4a displays the XPS full survey and spectra of C1s, N1s, Ce3d, and O1s, with binding energy values that align with previous research [30]. The C1s spectrum in Figure 4b displays peaks at 284.39 eV and 287.09 eV, which suggest the presence of carbon contaminants or adventitious carbon and sp^2^ hybridized carbon atoms, respectively [31]. The N1s spectrum in Figure 4c shows a main peak at 398.07 eV, which is attributed to sp^2^ hybridized aromatic C=N-C [32].

The Ce3d core-level spectrum in Figure 4d indicates the presence of seven coordination numbers of Ce^4+^ and reduced oxygen defects in a Ce^4+^-type fluorite arrangement. The coexistence of Ce^3+^/Ce^4+^ is confirmed through the deconvolution of the Ce 3d core-level spectrum. According to the literature, the 3d_3/2_ and 3d_5/2_ spin-orbit states are divided into two groups [33]. Peaks at 881.89, 888.29, 898.02, 902.39, 906.98, and 916.01 eV are allocated to Ce^4+^ states, while peaks at 884.77 and 900.6 eV belong to Ce^3+^ positions. The presence of Ce^3+^ positions is also indicated by the existence of O_2_ vacancies peaks in the O1s band, suggesting that the ceria surface is not fully oxidized and there may be an alteration in its electron structure [34]. The O1s core-level spectrum in Figure 4e reveals the presence of -O_2_ vacancies in Ce^3+^ and Ce^4+^ through peaks at 530.68, and 528.98 eV, respectively, supporting the coupling of CeQDs nanosized particles on CN sheets and the formation of the CeQDs-CN nanostructure [35].

The CV and GCD methods were employed to determine the electrochemical behavior of the created electrodes. The electrochemical performance of CN and CeQDs/CN-3 electrodes was first evaluated using CV in a three-electrode setup in a 1 mol/L KOH. The CV results were verified at various scan rates (1–20 mV/s) within a voltage range of 0–0.6 V vs. Hg/HgO (see Figure 5a,b). The voltammograms show clear redox peaks during the anodic and cathodic sweeps and their impact on the faradic pseudocapacitance. With cumulative sweep rate from 1 to 20 mV/s, the redox peak separation ΔE_redox_ increased from 1 to 7 mV due to electrode overpotential. Despite the high sweep rate of 20 mV/s, the redox peaks are still well resolved, indicating good rate capability for the CeQDs/CN-3 electrode material. The CeQDs/CN-3 electrode has greater CV area than the CN electrode due to the redox properties of CeQDs and CN. To compare the supercapacitive performance of CeQDs/CN-3, the CV curves of bare Ni-foam, CN, and CeQDs/CN-3 electrodes are compared at a potential of 0.6 V and 10 mV/s scan rate (Figure 5c). The current retort of the nickel foam is small, while the CeQDs/CN-3 electrode shows a larger CV curve area than the CN, implying higher electrochemical activity and considerable specific capacitance for the CeQDs/CN-3 electrode material. The larger CV curve area is due to more redox sites and increased exposure of these active sites. The incline of log i_p_ vs. log ν (Figure 5d) displays that the estimated b values for the anodic peaks of CN and CeQDs/CN-3 electrode materials are 0.64 and 0.67, respectively, indicating a combined pseudocapacitive response from capacitive and insertion (faradic) processes. It is worth noting that the CeQDs/CN-3 electrode displays a more dominant faradic contribution due to the presence of Ce sites in its framework. The capacitive contributions of CeQDs/CN-3 at sweep rates of 1, 2, 5, 10, and 20 mV/s are 81.1%, 85.85%, 90.56%, 93.13%, and 95.04%, respectively, showing an increase in capacitive effect with respect to the sweep rate (see Figure 5e). The increase in capacitive contribution is due to increased ion transport motion and shorter diffusion pathways. Figure 5f shows the diffusive and capacitive contributions of the CV profile at 5 mV/s, revealing that the diffusive contribution is primarily associated with the CeQDs/CN-3 electrode, due to the quantum effect of Ce ions and improved ion transport facilitated by interfacial contacts, leading to improved electrochemical performance.

Figure 6a,b show the GCD curves for CN and CeQDs/CN-3 electrodes at different current densities between 0.75–5 A/g. Figure 6c compares the GCD curves for bare Ni foam, CN, and CeQDs/CN-3 electrodes recorded at 0.75 A/g. The CeQDs/CN-3 electrode has a longer discharge time than the CN electrode, indicating a higher capacitance. The potential profiles of both electrodes display nonlinear GCD patterns resembling redox properties, demonstrating a battery-like behavior with quasi-reversible faradic reactions. Due to these reactions, the charge-to-voltage ratio changes over time. The longer discharge time of the CeQDs/CN-3 electrode is attributed to the redox properties of CeQDs ions, which aligns with the CV curves. The porous structure of the CeQDs/CN-3 electrode allows for faster ion transfer to the interior, enhancing its electrochemical performance with a quick I-V response. The specific capacitance (Cs) of the CeQDs/CN-3 electrode material at various constant discharge currents can be calculated using Equation (S1) based on the discharge curves.

Figure 6d displays the specific capacitance for both CN and CeQDs/CN-3 electrodes at different current densities. The CeQDs/CN-3 electrode has specific capacitance values of 202.5, 186.9, 164.52, 129.6, and 97.9 F/g at 0.75, 1.5, 3.0, 4.0, and 5 A/g, respectively. On the other hand, the CN electrode has specific capacitance values of 96.9, 84.96, 59.76, 37.12, and 22.7 F/g at 0.75, 1.5, 3.0, 4.0, and 5 A/g. The CeQDs/CN-3 electrode has an estimated specific capacitance that is approximately 2.08 times that of the CN electrode, due to the synergistic interaction between CeQDs and CN. This interaction enhances the rate capability, promoting quicker electron transportation and more capable ion diffusion into the composite material’s redox spots. The comparison confirms that the CeQDs/CN-3 electrode will yield higher current values. Appendix A Appendix A compares the g-C_3_N_4_-based electrode reports in existing published studies, showing that the CeQDs/CN-3 electrodes exhibit notable electrochemical performance.

With regard to the longevity of electrodes being a critical aspect in electrochemical devices, the cycling steadiness of the electrodes was assessed. Figure 6e showcases the capacitance retention and Coulombic efficiency (CE) of the CeQDs/CN-3 electrode after 10,000 cycles. This electrode proves to have excellent long-term stability with a capacitance holding of 95.4% after 10,000 sets at 5 A/g. The electrochemical reversibility is demonstrated by the GCD profiles shown in the inset, which exhibit good stability. The CeQDs/CN-3 electrode exhibits a CE of 99.38% in the first cycle and retains a CE of 99.89% even after 10,000 cycles.

Figure 6f shows the Nyquist plots (Z′ vs. −Z″) of the CN and CeQDs/CN-3 electrodes, show a combination of a depressed semicircle in the high-frequency region and a steep line in the low-frequency range. The reduced half-circle designates the R_ct_ resulting from faradic reactions, and the steep line signifies the capacitive behavior of the electrode (ideally, it would be a vertical line for an ideal capacitor). The R_s_ includes intrinsic resistance of the electrode material, bulk resistance of the electrolyte, and contact resistance at interfaces, can be determined from the real axis intercept in the high-frequency region. The R_ct_ values of the CN and CeQDs/CN-3 electrodes were calculated to be 13.81 Ω and 12.7 Ω, respectively, which demonstrate high charge transfer rates.

### Hybrid Coin Cell-Type Asymmetric Supercapacitor Device (HCASD)

Based on the outstanding electrochemical activity of the CeQDs/CN-3 electrode, a CeQDs/CN-3//AC HCASD with an alkaline PVA/KOH gel electrolyte was fabricated and evaluated. Figure 7a displays the CV profiles of the positive and negative electrodes, recorded at a sweep rate of 20 mV/s in a three-electrode configuration. The CeQDs/CN-3 was scanned in 0–0.6 V, while the AC electrode was scanned in −0.6 to 0 V. Figure 7b displays the CV profiles of the CeQDs/CN-3//AC HCASD at different applied potentials (0.4–1.2 V), indicating that 0–1.2 V is the optimal potential window for the HCASD. Figure 7c presents the CV profiles obtained at an enhanced working voltage of 1.2 V at 2–500 mV/s. The increase in CV curve area with scan rate shows that the asymmetric device has good electrochemical behavior. Furthermore, the GCD of the HCASD (Figure 7d) at 1.5–8 A/g demonstrate a high rate capability.

Figure 8a displays the graph of capacitance vs. current density for the HCASD, with estimated specific capacitances of 46.25, 26.57, 13.12, 9.6, and 5.7 F/g at 1.5, 3, 5, 6, and 8 A/g, correspondingly. The specific capacitance decreases linearly with an increase in current density, with a rate of 8.11%. Figure 8b shows the EIS spectrum of the HCASD along with a fitted curve, with estimated R_s_ and R_ct_ values of 2.35 Ω and 5.674 Ω, respectively. Figure 8c depicts the results of the stability and durability test of the HCASD over 10,000 cycles at 8 A/g. The test revealed that the CeQDs/CN-3//AC HCASD maintained 86.8% of its initial capacitance, indicating exceptional stability. The Ragone plot (Figure 8d) of the HCASD, which was obtained from the GCD curves using Equations (S2) and (S3), demonstrates an E_d_ of 9.25 Wh/kg at a P_d_ of 900 W/kg. The electrochemical characteristics of the fabricated HCASD noticeable the values reported in the literature [36,37]. Joseph et al. [36] showed through their study that a symmetrical device exhibits an exceptional E_d_ of 46 Wh/kg at P_d_ of 2691 W/kg. This remarkable performance is attributed to the binary materials’ interface, which facilitates a greater accessibility for ion transport. Wei et al. [37] demonstrated in their research that a symmetrical device can deliver an E_d_ of 10.4 Wh/kg, with a P_d_ of 187.3 W/kg by effectively combining the benefits of ZnS and g-C_3_N_4_ materials. These results highlight the promising potential of symmetrical devices in delivering high energy and power density for various applications. Figure 8e presents a 3D plot of energy vs. power density vs. time of discharge. The substantial energy-power densities and their long-standing stability are essential parameters for device applications. To validate the notable charge storage characteristics of the developed asymmetric supercapacitor device, two HCASDs were linked in sequence to illumination a red LED (Figure 8f), and it was able to function for 140 s. This test demonstrates the potential applications of the CeQDs/CN-3//AC supercapacitor for wearable electronics. The 2D/0D nanostructure created shows exceptional performance as a supercapacitor electrode, boasting a high capacitance and a remarkable rate capability and stability, surpassing other CN-based electrodes and the current advanced CN binary electrodes. The binary combination of materials results in an asymmetric device that delivers high energy and power densities, even after multiple cycles. This is due to the quantum structural features of CeO_2_QDs and layered CN, which provide more accessible sites for ion transport.

## 3. Experimental Section

### Preparation of g-C_3_N_4_/CeO_2_ Nanostructure

The synthesis of g-C_3_N_4_/CeO_2_ nanostructures was accomplished using a simple thermal decomposition method. Two grams of urea was decomposed at 450 °C to yield a yellow product denoted CN, as seen in Figure 9. CeO_2_ was synthesized by grinding 2 g of cerium (III) nitrate hexahydrate and subjecting it to a thermal treatment at 450 °C for 12 h, followed by washing with ethanol and water and drying. The resulting product was denoted CeQDs. The CeO_2_/g-C_3_N_4_ nanostructures were produced by grinding a mixture of urea and ceria precursors in the weight ratios of 1: 0.8, 1: 0.9, and 1: 1, and following the same procedure as for CeO_2_ synthesis. The final product was washed, dried overnight at 100 °C, and designated CeQDs/CN-1, CeQDs/CN-2, and CeQDs/CN-3. Further details on material characterization, electrochemical 3 electrode testing, and asymmetric device fabrication can be found in the Appendix A.

## 4. Conclusions

In conclusion, the synthesis of this novel 2D/0D CeQDs/CN nanostructure and its evaluation as an electrode for supercapacitors is of great scientific and technological significance. It delivers new perceptions into the electrochemical performance of CeQDs/CN nanostructures and opens up new opportunities for further research in this field. CN offers a great platform for growing CeO_2_ quantum dots (CeQDs), which are attractive materials for supercapacitor uses due to their large surface area, high reactivity, and high capacitance. This work is scientifically novel as it presents a simple synthesis approach for synthesizing CeQDs/CN nanostructures, which can guide the controlled growth of CeQDs. The 2D/0D nanostructure ensures good dispersion of CeQDs, thereby enhancing the electrode’s capacitance. Moreover, the results show that the CeQDs/CN nanostructure exhibits excellent rate capability and cycling stability, outperforming CN and CeQD counterparts and the current advanced reported CN binary electrode.

## Figures and Tables

**Figure 1 molecules-28-02489-f001:**
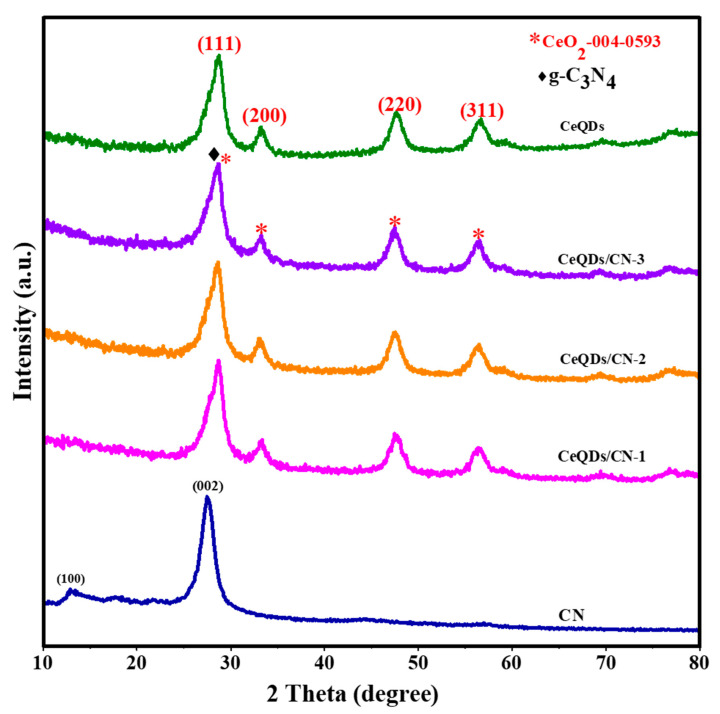
XRD pattern of the CN, CeQDs, CeQDs/CN-1, CeQDs/CN-2, and CeQDs/CN-3 nanostructures.

**Figure 2 molecules-28-02489-f002:**
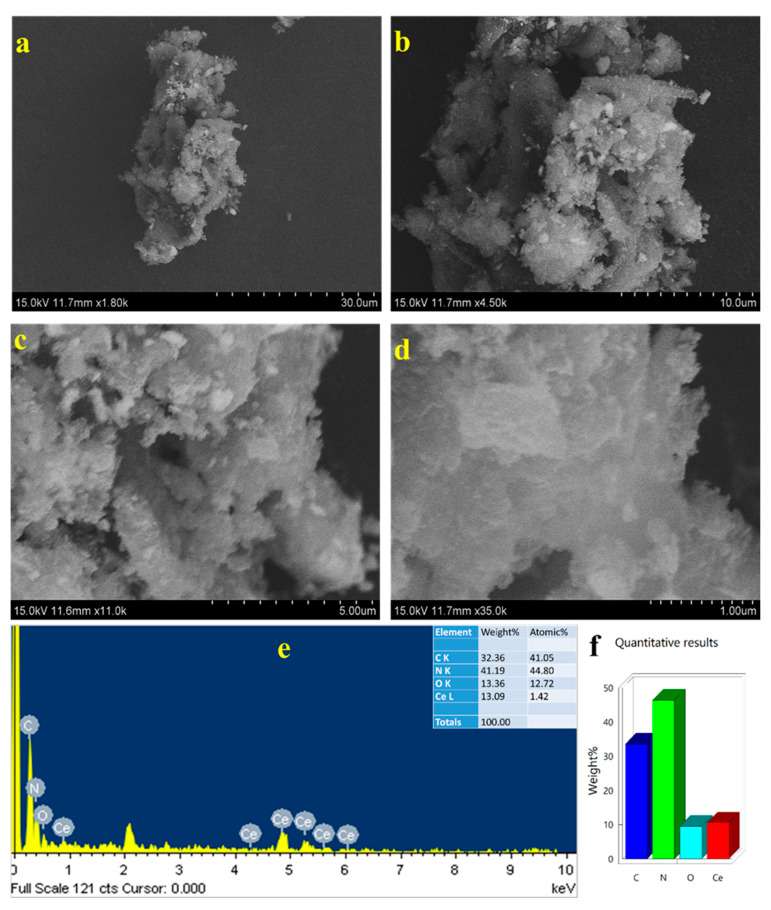
Different magnifications of FESEM images (**a**–**d**) and (**e**,**f**) EDX of CeQDs/CN-3 nanostructure.

**Figure 3 molecules-28-02489-f003:**
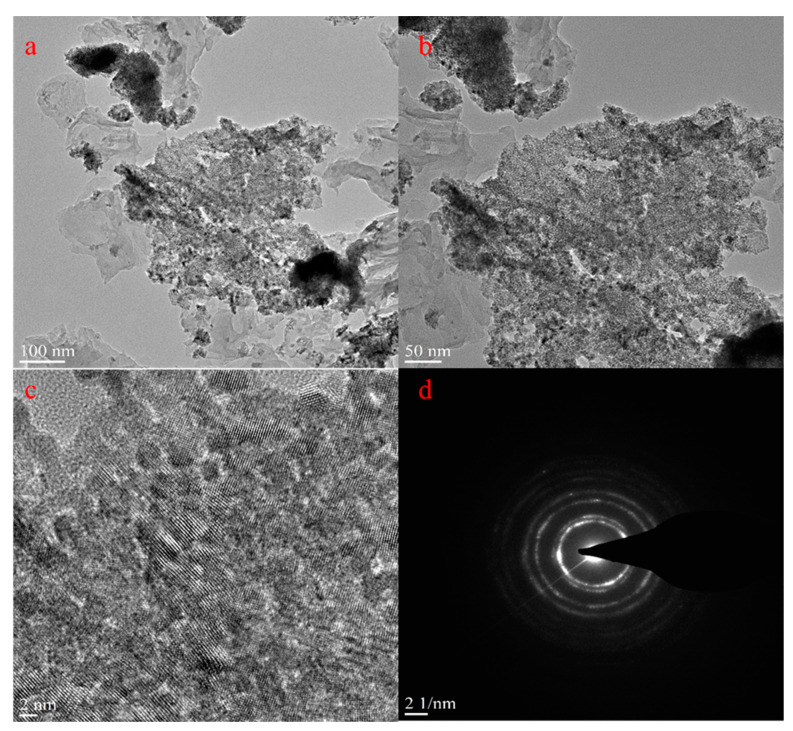
(**a**–**c**) HRTEM images and (**d**) SAED pattern of CN/CeQDs-3 nanostructure.

**Figure 4 molecules-28-02489-f004:**
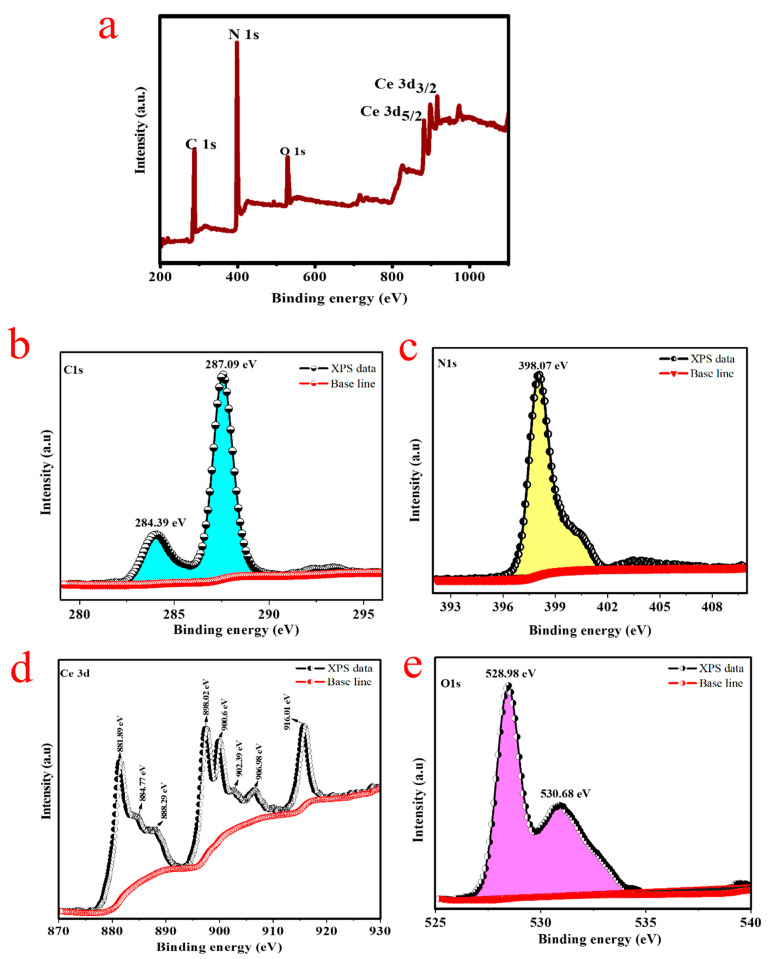
XPS results; (**a**) survey scan, (**b**) C1s, (**c**) N1s, (**d**) Ce3d and (**e**) O1s of CN/CeQDs-3 nanostructure.

**Figure 5 molecules-28-02489-f005:**
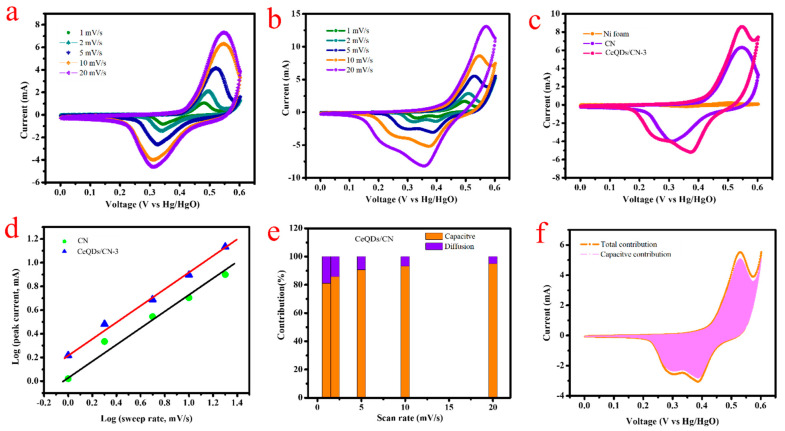
Three-electrode performance: CV curves of (**a**) CN, (**b**) CN/CeQDs-3, (**c**) comparison of CV curves of bare nickel foam, CN, and CN/CeQDs-3 at 10 mV/s, (**d**) analysis of b value of the cathodic and anodic peaks of CN/CeQDs-3 at different scan rates, (**e**) capacitive and diffusion contribution of CN/CeQDs-3, and (**f**) capacitive and diffusive contribution of CN/CeQDs-3 electrode at 5 mV/s.

**Figure 6 molecules-28-02489-f006:**
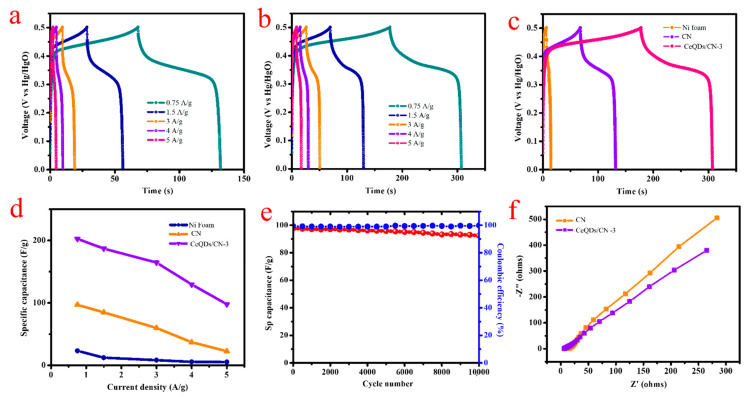
Charge/discharge curves of (**a**) CN and (**b**) CN/CeQDs-3, (**c**) comparison of charge discharge curves of bare nickel foam, CN and CN/CeQDs-3, (**d**) specific capacitance vs. current density of bare nickel foam, CN and CN/CeQDs-3, (**e**) cycling stability CN/CeQDs-3, and (**f**) EIS spectra of CN and CN/CeQDs-3 nanostructure.

**Figure 7 molecules-28-02489-f007:**
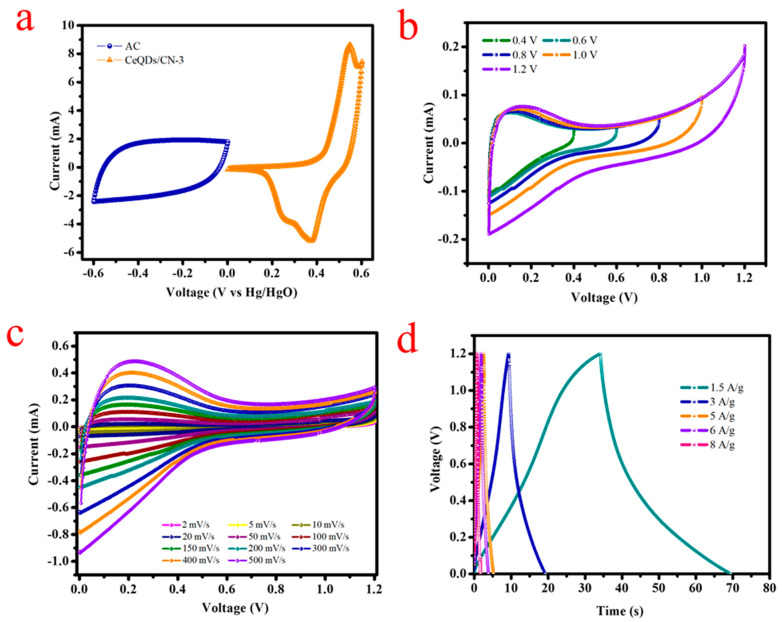
Electrochemical performance of the CN/CeQDs-3//AC HCASD device: (**a**) comparison of the CV curves of AC and CN/CeQDs-3 using three-electrode configuration at 20 mV/s; (**b**) CV curves of HCASD at different potentials; (**c**) CV profiles at different scan rates with a potential window of 1.2 V; (**d**) GCD profiles with different current densities of HCASD device.

**Figure 8 molecules-28-02489-f008:**
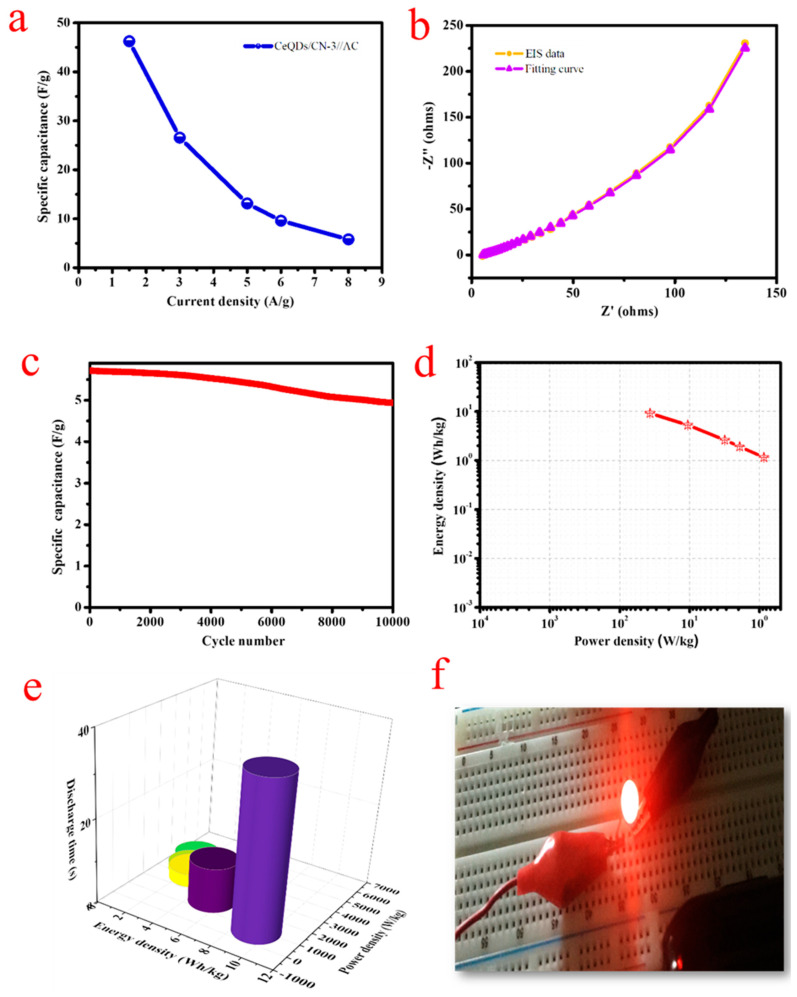
(**a**) Specific capacitance vs. current densities, (**b**) Nyquist plot with fitting curves of HCASD, (**c**) stability of the HCASD tested at 8 A/g, (**d**) Ragone plot, (**e**) 3D plot of energy density vs. power density vs. discharge time, and (**f**) practical applications of the CN/CeQDs-3//AC HCASD device: digital images of a red LED lit by the two HCASD.

**Figure 9 molecules-28-02489-f009:**
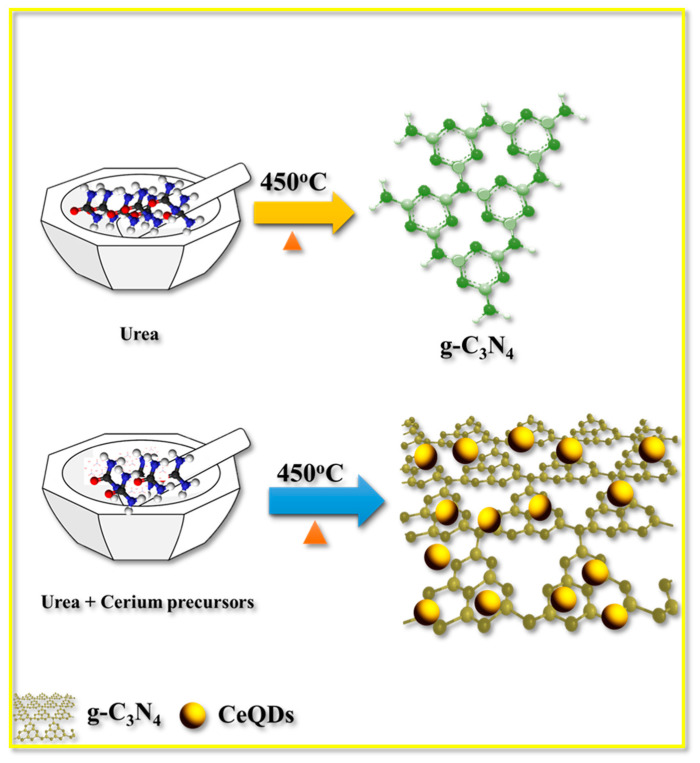
Schematic diagram of the CN and CeQDs/CN nanostructure synthesis process.

## Data Availability

All data are provided in this article.

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
