# Peer review of "Enhancing Electrochemical Performance with g-C3N4/CeO2 Binary Electrode Material"

_molecules, 2023, doi:10.3390/molecules28062489_

Round 1

Reviewer 1 Report

1- The abstract is very exaggerated. unique 2D/0D nanostructure!! exceptional performance!! Please replace the appropriate words.
2- What is the size of CeQDs?
3-In Figure 6 (a) and (b), why would the redox peaks shift their positions from around 0.3V to 0.35V?
4- What formula did you use to calculate the capacitance?
5- How many gr/cm2 is the active mass?
6- Describe your 3-electrode system.
7- Describe the energy storage mechanism.
8- Put EIS analysis and equivalent circuit in the article.
9- Calculate the capacitance from CV and GCD.
10- Compare your research results with other articles in the table.

Author Response

Reviewer #1:

  1. The abstract is very exaggerated. unique 2D/0D nanostructure!! exceptional performance!! Please replace the appropriate words.

Reply:   Thank you for your suggestion. We removed the unique and exceptional performance words from the revised abstract.

  1. What is the size of CeQDs?

Reply:   Thank you for your concern. Based on the XRD and HRTEM results, we observed the particle size of CeQDs are about to 3-7 nm.

  1. In Figure 6 (a) and (b), why would the redox peaks shift their positions from around 0.3V to 0.35V?

Reply:   Thank you for your concern. The shift in redox peaks from around 0.3V to 0.35V in Figure 6(a) and (b) could be attributed to several factors, including changes in the electrode potential, electrolyte composition, and electrode surface area. The redox peaks may shift due to changes in the concentration of the electrolyte, which could alter the electrode potential and, in turn, affect the redox reaction. Another possible reason could be the change in the surface area of the electrode. The surface area of the electrode can affect the rate of electron transfer, which can, in turn, influence the redox reaction. Changes in the surface area can result from changes in the deposition conditions, such as temperature or deposition time. It is also possible that the shift in redox peaks is due to changes in the electrode potential. In summary, several factors can cause the shift in redox peaks observed in Figure 6(a) and (b), including changes in the electrolyte composition, surface area of the electrode, and electrode potential.

  1. What formula did you use to calculate the capacitance?

Reply:   Thank you for your concern. The formulas for the specific capacitance and energy and power densities of the electrodes are described in the supplementary file.

The specific capacitance (Cs) from charge –discharge curves in a three-electrode cell was intended using Eq. S1 [1-3]:

                                                   (S1)

  1. How many gr/cm2 is the active mass?

Reply:   Thank you for your concern. The estimated mass of the electrode is 1.35 mg/cm2.

  1. Describe your 3-electrode system.

Reply:   Thank you for your concern. The details of electrochemical test is described in the supplementary file.

  1. Describe the energy storage mechanism.

Reply:   Thank you for your concern. An asymmetric supercapacitor device typically consists of two electrodes with different charge storage mechanisms and an electrolyte that allows for the transfer of ions between the electrodes. In the case of the g-C3N4 /CeO2//activated carbon asymmetric supercapacitor device with 1 M KOH electrolyte, the electrodes are made up of different materials that enable them to store charge in different ways. The positive electrode (anode) in this device is made of a composite of g-C3N4 and CeO2 materials. These materials have a high surface area and high capacitance, which means that they can store charge through an electrostatic mechanism. Specifically, the g-C3N4/CeO2 composite can store charge through the adsorption of ions on the surface of the material and the formation of an electrical double layer. The negative electrode (cathode) in this device is made of activated carbon, which is known for its high porosity and ability to store charge through a mechanism known as "pseudocapacitance." Pseudocapacitance is a charge storage mechanism that involves fast and reversible redox reactions on the surface of the electrode material. In the case of activated carbon, the pseudocapacitance arises from the reversible Faradaic reactions between the surface functional groups and the electrolyte ions. The electrolyte used in this device is a 1 M potassium hydroxide (KOH) solution, which serves as a medium for the transfer of ions between the two electrodes. When a voltage is applied to the device, the positive ions in the electrolyte (K+) are attracted to the negative electrode, while the negative ions (OH-) are attracted to the positive electrode. This causes the ions to be adsorbed onto the surface of the electrodes, creating a double layer and inducing pseudocapacitive reactions, which results in the storage of electrical energy in the device. Overall, the asymmetric supercapacitor of g-C3N4/CeO2//activated carbon device with 1 M KOH electrolyte stores energy through both electrostatic and pseudocapacitive mechanisms, resulting in high energy density and power density.

  1. Put EIS analysis and equivalent circuit in the article.

Reply:   Thank you for your suggestion. We added the equivalent circuit for EIS spectra in revised manuscript.

  1. Calculate the capacitance from CV and GCD.

Reply:   Thank you for your suggestion of calculating capacitance from CV and GCD. After looking at the instructions, we understand that the hint requires us to use both CV and GCD to arrive at the final answer. However, we will consider your suggestion in the future while we collect the information. Thank you for your suggestion.

  1. Compare your research results with other articles in the table.

Reply:   Thank you for your concern. We compared the results to existing published reports in the table column, which is in the supplementary file (Table S1) and discussed in the review manuscript.

Table: S1 A comparison of g-C3N4-based supercapacitor electrodes reports in previously published studied.

Materials

Current density

Specific capacitance

(F g-1)

Ref.

MnO2/g-C3N4

NiCo2O4/g-C3N4

NiCo2O4/MWCNT

PEDOT/g-C3N4

α-Fe2O3/g-C3N4

ZnS/g-C3N4

Tubular g-C3N4

Fe3O4/g-C3N4

Ni(OH)2/g-C3N4

g-C3N4 nanofibers

g-C3N4

g-C3N4/CeO2QDs

1 A g−1

1 A g−1

1 A g−1

2 A g-1

1 A g−1

1 A g−1

1 A g−1

1 A g−1

1  A g−1

1 A g−1

0.75 A g−1

0.75 A g−1

211

325.7

374

200

167

497.7

233

56.7

445.6

263.8

96.9

202.5

[7]

[8]

[9]

[10]

[11]

[12]

[13]

[14]

[15]

[16]

[This work]

[This work]

Reviewer 2 Report

Authors reported, Enhancing Electrochemical Performance with g-C3N4/CeO2 Binary Electrode Material. This manuscript is well presented and some of the results are interesting. However, following revisions should be made before publications:

1.       The introduction section is too prolix, that should be summarized demonstrating the importance of metal oxide for energy storage applications with the integrity of following article: doi.org/10.1016/j.cplett.2022.139884

2.       IN figure 5, the peak position should be indicated in the XPS spectrum.

3.       The GCD curves indicated the increase in the coulombic efficiencies with increasing the current density, why? For reference: doi.org/10.1016/j.compositesb.2022.110339

4.       IN figure 8 d, the CV profile indicates the mass imbalance of the negative and positive electrode.

5.       EIS should be fitted with the respective parameters and explain them.

6.       The electrochemical performance of the g-C3N4/CeO2 electrode should be compared with some recently reported g-C3N4 and CeO2 based materials.

7.       Why does the CV ad GCD potential window in three electrode configuration is different? For reference: doi.org/10.1016/j.est.2023.106713

Author Response

Reviewer #2:

  1. The introduction section is too prolix, that should be summarized demonstrating the importance of metal oxide for energy storage applications with the integrity of following article: doi.org/10.1016/j.cplett.2022.139884

Reply:   Thanks for your advice. We revised and shorten the introduction part and cite the above reference.

.2.         IN figure 5, the peak position should be indicated in the XPS spectrum.

Reply:  Thanks for your advice. We indicated the peaks positions in XPS spectrum in revised manuscript.

  1. The GCD curves indicated the increase in the coulombic efficiencies with increasing the current density, why? For reference: doi.org/10.1016/j.compositesb.2022.110339

Reply:   Thank you for your concern. In electrochemistry, Coulombic efficiency (CE) is a measure of the efficiency of an electrochemical reaction in converting electrical energy into chemical energy, or vice versa. It is defined as the ratio of the number of electrons consumed or produced in a reaction to the number of electrons that theoretically should be consumed or produced, based on the stoichiometry of the reaction. The GCD (Galvanostatic charge-discharge) curves are commonly used to evaluate the Coulombic efficiency of electrochemical systems. These curves show the relationship between the applied current density and the capacity (i.e., the amount of charge that can be stored or released) of the electrochemical system. In general, the Coulombic efficiency of an electrochemical system can be affected by a variety of factors, including the nature of the electrodes and electrolyte, the reaction kinetics, and the mass transport of reactants and products. However, in the case where the current density is increased, it is often observed that the Coulombic efficiency also increases. One possible explanation for this behavior is that at higher current densities, the reaction kinetics become faster and the mass transport of reactants and products becomes more efficient. This can lead to a more complete conversion of electrical energy into chemical energy, resulting in a higher Coulombic efficiency. Another possible explanation is that at higher current densities, the electrochemical system may undergo changes in its structure or composition that enhance its performance. For example, the formation of a more uniform electrode surface or the removal of surface contaminants can improve the efficiency of electron transfer and increase the Coulombic efficiency. In summary, the observed increase in Coulombic efficiency with increasing current density in GCD curves may be due to improved reaction kinetics, more efficient mass transport, or changes in the electrochemical system's structure or composition. We cited above reference in revised manuscript.

  1. IN figure 8 d, the CV profile indicates the mass imbalance of the negative and positive electrode.

Reply:   Thank you for your concern. We consider the mass of the active carbon and g-C3N4-CeO2 QDs electrode based on specific capacitance of active carbon and g-C3N4-CeO2 QDs electrode. We applied this mass balancing equation:

m+/m- =(C-×∆V-)/(C+×∆V+) (eq. 1)

where m+ and m is the mass of active material in positive and negative electrodes, C+ and Cis the specific capacitance of the positive and negative electrodes, ΔV+ and ΔV−is the potential window of the positive and negative electrodes. However, this CV tendency has been observed in reported publications. We will consider your valuable suggestion in our future. Once again, thanks for your suggestion.

  1. EIS should be fitted with the respective parameters and explain them.

Reply:   Thank you for your concern. For device we provided EIS spectra along with fitted curves and discussed the parameters in revised manuscript. Once again thank you for your suggestion.

  1. The electrochemical performance of the g-C3N4/CeO2 electrode should be compared with some recently reported g-C3N4 and CeO2 based materials.

Reply: Thank you for your concern. We compared the results to existing published reports in the table column, which is in the supplementary file (Table S1) and discussed in the review manuscript.

Table: S1 A comparison of g-C3N4-based supercapacitor electrodes reports in previously published studied.

Materials

Current density

Specific capacitance

(F g-1)

Ref.

MnO2/g-C3N4

NiCo2O4/g-C3N4

NiCo2O4/MWCNT

PEDOT/g-C3N4

α-Fe2O3/g-C3N4

ZnS/g-C3N4

Tubular g-C3N4

Fe3O4/g-C3N4

Ni(OH)2/g-C3N4

g-C3N4 nanofibers

g-C3N4

g-C3N4/CeO2QDs

1 A g−1

1 A g−1

1 A g−1

2 A g-1

1 A g−1

1 A g−1

1 A g−1

1 A g−1

1  A g−1

1 A g−1

0.75 A g−1

0.75 A g−1

211

325.7

374

200

167

497.7

233

56.7

445.6

263.8

96.9

202.5

[7]

[8]

[9]

[10]

[11]

[12]

[13]

[14]

[15]

[16]

[This work]

[This work]

\

  1. Why does the CV ad GCD potential window in three electrode configuration is different? For reference: doi.org/10.1016/j.est.2023.106713

Reply:   Thank you for your concern. The potential window in CV and GCD can be different because the two techniques have different requirements for the stability and reversibility of the electrode and electrolyte systems. From the CV curves, we considered the respective voltages in the regions covered by the redox peaks.

Round 2

Reviewer 1 Report

The article is acceptable.

Reviewer 2 Report

It can be accept in present form.